# Phytochemical-Assisted Synthesis of Fe_3_O_4_ Nanoparticles and Evaluation of Their Catalytic Activity

**DOI:** 10.3390/mi13122077

**Published:** 2022-11-26

**Authors:** Rokeya Khatun, Muhammad Shamim Al Mamun, Suravi Islam, Nazia Khatun, Mahmuda Hakim, Muhammad Sarwar Hossain, Palash Kumar Dhar, Hasi Rani Barai

**Affiliations:** 1Chemistry Discipline, Khulna University, Khulna 9208, Bangladesh; 2Industrial Physics Division, Bangladesh Council of Scientific and Industrial Research, Dhaka 1205, Bangladesh; 3Biomedical and Toxicological Research Institute (BTRI), Bangladesh Council of Scientific and Industrial Research, Dhaka 1205, Bangladesh; 4Department of Chemistry, Sogang University, Seoul 04107, Republic of Korea; 5Department of Mechanical Engineering, Yeungnam University, Gyeongsan 38541, Republic of Korea

**Keywords:** green synthesis, Fe_3_O_4_ NPs, *Baccaurea ramiflora*, catalytic degradation, ecofriendly

## Abstract

In this study, magnetite nanoparticles (Fe_3_O_4_ NPs) were synthesized using *Baccaurea ramiflora* leaf extracts and characterized by visual observation, UV–Vis, FTIR, XRD, FESEM, and EDS. The UV−Vis spectrum showed continuous absorption at 300–500 nm, confirming the formation of Fe_3_O_4_ NPs. FTIR revealed that compounds containing the O-H group act as reducing agents during Fe_3_O_4_ NPs formation. Agglomerated spherical NPs were observed in the FESEM image. The prominent peak at ~6.4 keV in the EDS spectrum ascertained the existence of Fe, while the sharp peak at ~0.53 keV confirmed the presence of elemental oxygen. XRD patterns affirmed the crystalline nature. The size of as-synthesized NPs was observed to be 8.83 nm. The catalytic activity of Fe_3_O_4_ NPs for the reduction of methylene blue (MB) dye was monitored by UV–Vis. The maximum absorption peak of MB dye at 664 nm was almost diminished within 20 min, which revealed Fe_3_O_4_ NPs could be an excellent catalyst for wastewater treatment.

## 1. Introduction

Nowadays, the release of organic dyes in water from industrial plants has become a major threat to mankind and also to the environment. Organic dyes have been utilized in the ventures like cosmetic, leather, food, textile, paper, and drug businesses for a long period of time. Particularly, azo dyes have been recognized as plausible carcinogenic agents because of their high stability and complex chemical structure, as they are not easily biodegradable. Released dyes are exceptionally impervious to microorganisms, so reduction through natural processes is required. A plethora of literature reported that NPs have vivid catalytic activity because of their large surface-to-volume ratios, and thus they can be used for the degradation of dyes [1,2,3].

Besides, the synthesis of NPs has become a matter of great interest due to their catalytic, optical, electrical, mechanical, and magnetic properties. Numerous techniques such as the sol–gel process [4], co-precipitation [5] sonochemical method [6], hydrothermal techniques [7], non-aqueous synthesis [8], ultrasound irradiation [9], thermal reduction [10], microemulsion method [11], etc. have been developed to synthesize NPs. However, conventional chemicals and physical methods increase environmental and biological hazards due to toxic reducing agents used during the synthesis procedures [3]. Meanwhile, green chemistry or biological synthesis has received much attention on account of its economic utilization of time and minimal usage of hazardous reducing chemicals [12]. However, the microbe-mediated synthesis of metallic NPs is not viable in an industrial setting because of the necessity of highly sterile conditions and their maintenance. Therefore, the application of plant extracts for the synthesis of NPs is potentially more convenient instead of the usage of microorganisms. The green route provides natural capping agents for the stabilization of metal nanoparticles without them being contaminated with hazardous chemicals [1,13,14].

By using the green method, different types of NPs such as magnetite [15], silver [16], copper [17], copper oxide [18], nickel oxide [19], zinc oxide [20], manganese dioxide [21], gold [22], etc. have been synthesized. Among them, Fe_3_O_4_ NPs are more significant and viable on account of their preeminent properties, for example, being superparamagnetic, biodegradable, biocompatible, and expected to be non-harmful to living organisms [1,23]. These extraordinary properties permit Fe_3_O_4_ NPs to be generally utilized in various regions of utilization, for example, in biosensors, catalysis, magnetic resonance imaging, magnetic storage media, and targeted drug delivery [15]. A study of various portions of the literature recommends that extracts from different parts of a plant, for example *Sargassum muticum* (seaweed) [24], *Kappaphycus alvarezii* (seaweed) [15], *Azadirachta indica* [25], Banana leaves [26], *Glycosmis mauritana* [27], *Zanthoxylum armatum* [28], *Cynara cardunculus* [29], *Ocinum sanctum* [30], *Centella asiatica* [31], *Punica granatum* [32], or *Gratophyllum pictum* [33], be exploited to synthesize Fe_3_O_4_ NPs. So far, green synthesis of Fe_3_O_4_ NPs using *B. ramiflora* leaves extract has not been performed yet. 

*B. ramiflora*, locally known as Lotkon, is a slow-growing evergreen tree in the Phyllanthaceae family with a dispersal crown and thin bark. It is found throughout Asia, and is generally cultivated in India, Bangladesh, and Malaysia. Different parts of *B. ramiflora* have potential medicinal value and are used in the treatment of skin diseases. Besides, this plant is well known due to its excellent cytotoxic, antioxidant, analgesic, neuropharmacological, anti-inflammatory, and antidiarrheal properties. The performance of a phytochemical assay of *B. ramiflora* plants showed the presence of carbohydrates, phenol, alkaloids, glycosides, flavonoids, terpenoids, sterol, resins, tannins, fixed oils, etc. [34,35,36]. It was assumed that the leaves of *B. ramiflora* could be used for the reduction of metal compounds as an effective reducing, stabilizing, and capping agent. Therefore, this study aimed to synthesize Fe_3_O_4_ NPs using *B. ramiflora* leaves and evaluate their catalytic activity. 

## 2. Experimental Methods

### 2.1. Extraction of Phytochemicals

The fresh leaves of *B. ramiflora* were collected from the local market of Khulna, Bangladesh. Approximately 100 g of fresh leaves was washed thoroughly several times with deionized water. This material was then shade-dried and made into fine powder form (~14 g) using a blender. About 5 g of *B. ramiflora* leaf powder was added to 100 mL of deionized water, and the mixture was boiled for 30 min at 70 °C. After that, the infusion obtained was allowed to cool down at room temperature and filtration was performed with Whatman filter paper No. 42. For further use, ~97 mL of the extract was preserved in a refrigerator at 4 °C [37]. 

### 2.2. Green Synthesis & Characterization of Fe_3_O_4_ NPs

Ferrous sulfate heptahydrate (FeSO_4_.7H_2_O) (Merck, Darmstadt, Germany) and Ferric Chloride hexahydrate (FeCl_3_.6H_2_O) (Loba Chemie, Maharashtra, India) were taken in a 1:2 molar ratio and dissolved in 50 mL deionized water. This solution was heated at 80 °C under mild stirring using a magnetic stirrer for 10 min. Then, 10 mL of plant extract was added slowly into the solution. After 5 min, freshly prepared 1.0 M NaOH (Loba Chemie, Maharashtra, India) was added into the solution drop by drop until uniform precipitation of magnetite nanoparticles was obtained. At that time, the pH of the solution became 11. The solution was kept undisturbed and allowed to cool down to room temperature. The black-colored nanoparticles were deposited at the bottom. The separation of deposited precipitate (Fe_3_O_4_ NPs) was conducted by a ring magnet, and decantation was performed to remove the unwanted foreign particles. The obtained product was finally dried at 70 °C in an oven [38]. It is noteworthy that the overall protocol was repeated as a control reaction without adding *B. ramiflora* leaves extract to the iron salt solution. However, to know the size, shape, surface texture, and chemical constituents, the synthesized magnetite nanoparticles were assessed for further characterization. Initially, the Fe_3_O_4_ NPs formation was confirmed via the magnetic bar and UV−Vis spectrophotometer (JASCO V−730, Tokyo, Japan). The existence of various biomolecules capped on the surface of nanoparticles was investigated by Fourier transform infrared (FTIR) spectroscopy (SHIMADZU, 8201 PC). The surface morphology and elemental composition were inspected by using a Philips X’PERT PRO X−Ray diffractometer and Field Emission Scanning Electron Microscopy (FEJSM−7600F, Tokyo, Japan), respectively.

### 2.3. Catalytic Reduction of MB Dye

The reduction of MB dyes using sodium borohydride (Merck, Darmstadt, Germany) in the presence and absence of Fe_3_O_4_ NPs was carried out to determine the catalytic activity of Fe_3_O_4_ NPs. An aqueous solution of 10 mL 0.01 M of NaBH_4_ and 10 mL 0.1 mM of MB was mixed in a beaker. After that, a sufficient amount of Fe_3_O_4_ NPs (0.4 g/L) was added separately into the solution, and the UV−Vis spectra of dye degradation were recorded at regular intervals of time.

## 3. Results and Discussion

### 3.1. Formation of Fe_3_O_4_ NPs

The formation of Fe_3_O_4_ NPs was initially assessed by (a) visual observation, (b) magnetic behavior, and (c) UV−Vis spectroscopy. The color of the reaction mixture was changed from dark brown to black at the time of synthesis and it was the key sign of the formation of Fe_3_O_4_ NPs [29]. Furthermore, the synthesized NPs in the aqueous solution were attracted to the ferrite ring magnet and deposited at the side of the magnet (Figure 1). When the ring magnet was rotated, the deposited Fe_3_O_4_ NPs were also rotated. This phenomenon was not observed without the external magnetic field, which specified the formation of Fe_3_O_4_ NPs. The chemical reaction of Fe_3_O_4_ NPs formation can be hypothesized by Equations (1) and (2) [12]. In the first step, it was presumed that the phytochemicals (polyphenols, flavonoids, polycarboxylic acid, etc.), present in the *B. ramiflora* leaf extract, acted as chelators to form the iron-phytochemicals complex. The next step was the conversion of phytochemicals-Fe(OH)_2_/Fe(OH)_3_ into spherical-shaped Fe_3_O_4_ NPs due to the addition of NaOH. In a similar study, Yew et al. [15] described that the positively charged Fe_3_O_4_ NPs are surrounded by the negatively charged groups of phytochemicals via the weak van der Waals forces that stabilize the molecular structure of magnetic particles.
(1)$$\mathrm{Plant}\;\mathrm{extract}+{\mathrm{Fe}}^{2+}\left(aq\right)+{\mathrm{Fe}}^{3+}\left(aq\right)+H_{2}O\left(l\right)\underset{\mathrm{Stirring}}{\to }\left[\mathrm{Phytochemicals}-{\mathrm{Fe}}^{2+}/{\mathrm{Fe}}^{3+}\right]\left(aq\right)$$
(2)$$\left[\mathrm{Phytochemicals}-\frac{{\mathrm{Fe}}^{2+}}{{\mathrm{Fe}}^{3+}}\right]\left(aq\right)+8{\mathrm{OH}}^{-}\left(aq\right)\underset{\mathrm{Stirring},\;{}^{\infty }∆}{\to }\;\left[\mathrm{Phytochemicals}-{\mathrm{Fe}}_{3}O_{4}\right]\left(s\right)↓+4H_{2}O\left(aq\right)$$

### 3.2. UV−Vis Spectral Analysis

Figure 2 shows the UV−Vis spectrum of Fe^2+^/Fe^3+^ (1:2), control (Fe^2+^/Fe^3+^ + NaOH), *B. ramiflora* leaves extract, and synthesized Fe_3_O_4_ NPs. The aqueous plant extract exhibited characteristic peaks at 272 nm and around 213 nm due to the presence of various phytochemicals [12]. The spectrum of only Fe^2+^/Fe^3+^ solution (black line) showed a broad peak around 300 nm, and the peak completely disappeared when NaOH solution was added to the salt solutions (control reaction, red line). On the other hand, the synthesized nanoparticles showed continuous absorption in the visible range of 300–500 nm without any strong absorption peak (green line) compared to the plant extract (blue line), aqueous Fe^2+^/Fe^3+^ (1:2) solution, and control reaction. This phenomenon probably occurred due to the reaction between iron salts and phytochemicals of plant extracts in the presence of NaOH solution. However, studies from Dhar et al. [12] and Yew et al. [15] stated identical UV−Vis spectra for Fe_3_O_4_ NPs synthesized using *Lathyrus sativus* peel extracts and seaweed extracts, respectively. By comparison, it was realized that the synthesized nanoparticles were Fe_3_O_4_ NPs.

### 3.3. FTIR Analysis

FTIR spectroscopy is an effective tool for the characterization of various functional groups that are capped on the surface of metallic NPs [15]. The FTIR spectrum of *B. ramiflora* leaves and the synthesized magnetite nanoparticles (Figure 3) exhibited characteristic bands at 3333 cm^−1^, 2335 cm^−1^, 1647 cm^−1^, 1414 cm^−1^, 1121 cm^−1^, 973 cm^−1,^ and 703 cm^−1^. The peak at 3333 cm^−1^ represents the existence of the O−H group of phenolic compounds, which mainly act as reducing agents for the NPs synthesis [28]. The band at 1647 cm^−1^ is attributed to the presence of the C=O group of carboxylic acid, which acts as a capping agent. The bands at 1414 cm^−1^ and 1121 cm^−1^ are attributed to the occurrence of aromatic amine (N−H bond) [29] and the C−O stretching frequency of phenolic compound, respectively [15,26]. The peak 703 cm^−1^ reveals the presence of an aromatic C−H bending band. The shifting of all FTIR bands denotes the formation of nanoparticles with the *B. ramiflora* extract.

### 3.4. FESEM and EDS Analyses

Figure 4a depicts the SEM images of Fe_3_O_4_ NPs, which were synthesized from aqueous leaf extract of *Baccurea ramiflora*. The observed morphology is semi-spherical with several agglomerates. These agglomerates are the results of the stearic effect attributed to the interactivities of active sites on the NPs’ surfaces. Yet, the magnetic interactions generated by the individual Fe_3_O_4_ NPs are also important in explaining these observed agglomerations [29]. The elemental analysis of Fe_3_O_4_ NPs is represented in Figure 4b. In the EDS image, a prominent peak at ~6.4 keV affirmed the presence of elemental iron (Fe), while an intense peak at ~0.53 keV ensured the existence of elemental oxygen (O). Similar types of results were previously reported by Dhar et al. [12]; Sirdeshpande et al. [14]; and Groiss et al. [13] while synthesizing Fe_3_O_4_ NPs using the *Lathyrus sativus* peel extract, and the leaf extract of *Calliandra haematocephala* and *Cynometra ramiflora*, respectively. However, the distinctive peak of iron in the range between 6 keV and 7 keV and oxygen in the EDS spectrum confirmed the formation of Fe_3_O_4_ NPs via the green approach.

### 3.5. XRD Analysis

The confirmation of the crystalline nature of Fe_3_O_4_ NPs was established using powder XRD. The XRD pattern of Fe_3_O_4_ NPs, synthesized using aqueous leaves of *B. ramiflora* leaves, is given below in Figure 5. The brag reflection peaks were detected at 2θ values at 30.1°, 35.5°, 43.21°, 57.01°, and 62.61°, respectively indexed to (220), (311), (400), (511) and (440) planes which exactly matched with JCPDS No. 19−0629. The size of MNPs was calculated using the Debye–Scherrer formula:(3)D=Kλ/βcosθ
where *D* is the crystalline domain size perpendicular to the reflecting planes, *K* is a shape factor (0.9), *λ* is the X-ray wavelength (0.1546 nm), *β* is the full width at half maximum (radian) and θ is the diffraction angle (radian) [12,15]. 

By applying Equation (3), the approximated crystalline size of NPs was 8.83 nm, which was calculated from the full-width half-maximum of Fe_3_O_4_ at a 311 plane and a diffraction peak at 2θ = 35.50°. The finding of this study was consistent with the mean size of 17.72 nm, 16.79 nm, and 18.69 nm observed by Dhar et al. [12], Yew et al. [15], and Kumar et al. [1], respectively. Equivalently, 13.5 nm, 20.7 nm, and 23.82 nm were also documented for the Fe_3_O_4_ NPs synthesized using *Cynara cardunculus* leaf extracts [29], the leaf extract of *Zanthoxylum armatum* DC [28], and *Peltophorum pterocarpum pod* extract [39], respectively. The Miller indices for the 311 planes were estimated based on Equations (4) and (5).
(4)$$d_{hkl}=\frac{\lambda }{2\sin\theta }$$
(5)$$a=d_{hkl}\times \sqrt{h^{2}+k^{2}+l^{2}}$$

The interplanar spacing (*d_hkl_*) and lattice parameters were observed as 2.536 Å and 8.409 Å, which were consistent with the previously reported magnetic standards (*d_hkl_* = 2.535 Å, a = 8.322) [40]. Furthermore, this data was in good agreement with the lattice parameter value of 8.377 Å [12], 8.4343 Å [39], and 8.399 Å [14], respectively for the Fe_3_O_4_ NPs (Table 1). 

### 3.6. Reduction of MB Dye

The usage of metal nanoparticles as a catalyst can be effective for the reduction of dyes as they have high reactive activity and specific surface area. The catalytic performance of Fe_3_O_4_ NPs was studied using MB dye. Figure 6a depicts the UV−Vis spectra of MB dye in the presence of only NaBH_4_. Interestingly, the absorbance and color of the dye remained unchanged even after 1 h. The effective depletion of MB dye with NaBH_4_ is observed in the presence of Fe_3_O_4_ NPs (catalyst) shown in Figure 6b. The maximum absorption of MB dye at 664 nm is gradually reduced over time, and adverts the catalyst which is to be used to deplete the MB dye. During the degradation process (Figure 1), the color of the MB dye solution became almost faded after 20 min. 

To estimate the rate constant of MB catalytic reduction by Fe_3_O_4_ NPs, a kinetic study was conducted. It is well known that the catalytic reduction of dye molecules follows a pseudo-first-order reaction [41]. Considering that reaction order, ln (A/A_0_) vs. time plot was plotted in Figure 6c. The slope of the fitted line (red) reflects the rate constant of the MB reduction. The rate constant for the MB was estimated to be 0.067 ± 0.010 min^−1^.

However, the catalytic efficiency of synthesized Fe_3_O_4_ NPs for the reduction of MB dyes has been compared with previously published similar reports (Table 2), something which complies with our current study. Overall, the phytochemicals present in the *B. ramiflora leaves* extracts could be used as a capping and stabilizing agent for the synthesis of spherical-shaped Fe_3_O_4_ NPs, which could further be used as potential catalysts for the reduction of MB dyes via an easy and eco-friendly approach.

## 4. Conclusions

The preparation of Fe_3_O_4_ NPs by *B. ramiflora* plants was performed based on the green method. The synthesized nanoparticles were initially confirmed by visual observation and the use of a UV−Vis Spectrophotometer. The formation of Fe_3_O_4_ NPs was further confirmed by the presence of noticeable absorption peaks of FTIR, and the elemental composition of iron and oxygen in the EDS spectrum. FESEM analysis indicated that most of the particles were agglomerated and spherically shaped. The magnetite structure was completely recognized by the XRD and the size of the crystal was estimated to be 8.83 nm. So, it can be concluded that the synthesis conditions were sufficient for the procurement of Fe_3_O_4_ NPs. Nevertheless, the green synthesis by *B. ramiflora* leaves extract offers a non-toxic and eco-friendly alternative for the preparation of Fe_3_O_4_ NPs. Furthermore, the prepared Fe_3_O_4_ NPs showed compatible catalytic activity in the methylene blue dye degradation process. The color of the MB dye solution became almost faded within 20 min and the rate constant was 0.067 ± 0.010 min^−1^. These findings signify the utility of Fe_3_O_4_ NPs as a proficient catalyst in wastewater management.

## Data Availability

It is available on requesting to corresponding authors.

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
