# Peer review of "Phytochemical-Assisted Synthesis of Fe3O4 Nanoparticles and Evaluation of Their Catalytic Activity"

_micromachines, 2022, doi:10.3390/mi13122077_

Round 1
Reviewer 1 Report
1. Since the powder from BR leaves are not a standard chemical. More information could be provided. In section 2.1, it would be better to note how many BR leaves are approximately needed to get 5g powder and how much extract can be obtained after the filtration.
2. The status note in equations 1 and 2 should be united, for Fe2+ (aq).
3. Please renumber the equations.
4. More information, especially the reaction performance data could be added in the Conclusion part.
Author Response
Reply to Reviewer #1
Comments and Suggestions for Authors
Comment-1. Since the powder from BR leaves are not a standard chemical. More information could be provided. In section 2.1, it would be better to note how many BR leaves are approximately needed to get 5g powder and how much extract can be obtained after the filtration.
Reply-1: Thank you so much for your valuable comments. We have accepted it and provided the required information in section 2.1. The fresh leaves of B. ramiflora were collected from the local market of Khulna, Bangladesh. Approximately 100g of fresh leaves were washed thoroughly several times with deionized water. These were then shade-dried and made into fine powder form (~14g) by using a blender. About 5g of B. ramiflora leaf powder was added to 100 mL of deionized water and the mixture was boiled for 30 minutes at 70 °C. After that, the infusion obtained was allowed to cool down at room temperature and filtration was done with Whatman filter paper No. 42. For further use, ~97 mL of extract was preserved in a refrigerator at 4 °C [37].
Comment-2. The status note in equations 1 and 2 should be united, for Fe2+ (aq).
Reply-2: Accepted and corrected in the revised manuscript:
Comment-3. Please renumber the equations.
Reply-3: Accepted and corrected.
Comment-4. More information, especially the reaction performance data could be added in the Conclusion part.
Reply-4: Accepted and more information regarding obtained results have been added. The preparation of Fe3O4 NPs by B. ramiflora plants was performed based on the green method. The synthesized nanoparticles were initially confirmed by visual observation and UV- Visible Spectrophotometer. The formation of Fe3O4 NPs was further confirmed by the presence of noticeable absorption peaks of FTIR, and the elemental composition of iron and oxygen in the EDS spectrum. FESEM analysis indicated that most of the particles were agglomerated spherical shaped. The magnetite structure was completely recognized by the XRD and the size of the crystal was estimated to be 8.83 nm. So, this can be concluded that the synthesis conditions were sufficient for the procurement of Fe3O4 NPs. Nevertheless, the green synthesis by B. ramiflora leaves extract offers a non-toxic and eco-friendly alternative for the preparation of Fe3O4 NPs. Furthermore, the prepared Fe3O4 NPs showed compatible catalytic activity in the methylene blue dye degradation process. The color of the MB dye solution became almost faded within 20 min and the rate constant was 0.067 min-1, which signifies the utility of Fe3O4 NPs as catalysts in wastewater management.

Reviewer 2 Report
The authors present a synthesis of iron oxide nanoparticles in the presence of plant extract that are included, minimally on the surface, of the particles. The synthesis is well described, and the particles are fully and convincing characterized. The authors provide evidence for the inclusion of organics on the particles as well, particularly through FTIR. These particles are used as catalysts in the reduction of methylene blue with sodium borohydride. Good activity is seen with a measured rate constant. This work is suitable for publication with one major point and a few minor points.
Major point: It is unclear that the organics affect the chemistry of the nanoparticles. A control reaction with a similarly prepared particle not in the plant extract should be included. There is some literature data, but it appears that conditions for the reduction between this and those studies are slightly different.
Minor points:
1. The experimental section does not provide a reproducible amount of nanoparticle in the description of the methylene blue reduction.
2. The rate constant seems like it has too many significant figures. Regardless of the sig figs, it should be reported with an error.
3. Table 2 should include the data from this work for easy comparison, even if not all data is present.
Author Response
Reply to Reviewer #2
Comments and Suggestions for Authors
The authors present a synthesis of iron oxide nanoparticles in the presence of plant extract that are included, minimally on the surface, of the particles. The synthesis is well described, and the particles are fully and convincing characterized. The authors provide evidence for the inclusion of organics on the particles as well, particularly through FTIR. These particles are used as catalysts in the reduction of methylene blue with sodium borohydride. Good activity is seen with a measured rate constant. This work is suitable for publication with one major point and a few minor points.
Major point:
Comment-1: It is unclear that the organics affect the chemistry of the nanoparticles. A control reaction with a similarly prepared particle not in the plant extract should be included. There is some literature data, but it appears that conditions for the reduction between this and those studies are slightly different.
Reply-1: Accepted. Thank you so much for your scholastic comment. To understand the effect of plant extract, we have added and explained the UV-Vis spectrum of (i) only Fe2+/Fe3+ solution and (ii) Fe2+/Fe3+ solution + NaOH solution as a control reaction (Section 3.2).
3.2. UV-Visible spectral analysis
Fig. 2 shows the UV-Visible spectrum of Fe2+/Fe3+ (1:2), control (Fe2+/Fe3+ + NaOH), B. ramiflora leaves extract, and synthesized Fe3O4 NPs. The aqueous plant extract exhibited characteristic peaks at 272 nm and around 213 nm due to the presence of various phytochemicals [12]. The spectrum of only Fe2+/Fe3+ solution (black line) showed a broad peak around 300 nm, and the peak completely disappeared when NaOH solution was added to the salt solutions (control reaction, red line). On the other hand, the synthesized nanoparticles showed continuous absorption in the visible range of 300-500 nm without any strong absorption peak (green line) compared to the plant extract (blue line), aqueous Fe2+/Fe3+ (1:2) solution, and control reaction. This phenomenon probably occurred due to the reaction between iron salts and phytochemicals of plant extracts in presence of NaOH solution. However, studies from Dhar et al. [12] and Yew et al. [15] stated identical UV-Visible spectra for Fe3O4 NPs synthesized using Lathyrus sativus peel extracts and seaweed extracts, respectively. By comparison, it was realized that the synthesized nanoparticles were Fe3O4 NPs.
Figure 2. UV- Visible spectrum of Fe2+/Fe3+ (1:2), control, B. ramiflora leaves extract and synthesized Fe3O4 NPs.
Minor points:
Comment-1. The experimental section does not provide a reproducible amount of nanoparticle in the description of the methylene blue reduction.
Reply-1: Accepted and provided (Line no.: 113, section 2.3).
2.3. Catalytic reduction of MB dye
The reduction of MB dyes using sodium borohydride (Merck, Germany) in the presence and absence of Fe3O4 NPs was carried out to determine the catalytic activity of Fe3O4 NPs. An aqueous solution of 10 mL 0.01 M of NaBH4 and 10 mL 0.1 mM of MB was mixed in a beaker. After that, a sufficient amount of Fe3O4 NPs (0.4 g/L) was added separately into the solution, and the UV-Vis spectra of dye degradation were recorded at regular intervals of time.
Comment-2. The rate constant seems like it has too many significant figures. Regardless of the sig figs, it should be reported with an error.
Reply-2: Thank you so much for your observation. Therefore, we have kept only a 3-digit significant figure with the error. The rate constant for the MB was estimated to be 0.067 ±0.010 min-1.
Comment-3. Table 2 should include the data from this work for easy comparison, even if not all data is present.
Reply-3: Accepted and provided in Table 2 in the revised manuscript.

Round 2
Reviewer 2 Report
Excellent and thoughtful revision has been made by the authors. I fully support publication of this manuscript.